# Research Advances of Purple Sweet Potato Anthocyanins: Extraction, Identification, Stability, Bioactivity, Application, and Biotransformation

**DOI:** 10.3390/molecules24213816

**Published:** 2019-10-23

**Authors:** Aoran Li, Ruoshi Xiao, Sijia He, Xiaoyu An, Yi He, Chengtao Wang, Sheng Yin, Bin Wang, Xuewei Shi, Jingren He

**Affiliations:** 1College of Food Science and Engineering, Wuhan Polytechnic University, Wuhan 430023, China; liaoran618@outlook.com (A.L.); rosalindxiao124@163.com (R.X.); 13377936777@163.com (S.H.); axy11_15@163.com (X.A.); 2Key Laboratory for Deep Processing of Major Grain and Oil, Ministry of Education, Wuhan 430023, China; 3Hubei Key Laboratory for Processing and Transformation of Agricultural Products, Wuhan Polytechnic University, Wuhan 430023, China; 4Beijing Engineering and Technology Research Center of Food Additives, Beijing; Technology & Business University (BTBU), Beijing 100048, China; yinsheng@btbu.edu.cn; 5Food College, Shihezi University, Shihezi 832000, China; binwang0228@shzu.edu.cn

**Keywords:** purple sweet potato anthocyanin, separation and extraction, structural characterization, stability, functional activity, biotransformation

## Abstract

Purple sweet potato anthocyanins are kinds of natural anthocyanin red pigments extracted from the root or stem of purple sweet potato. They are stable and have the functions of anti-oxidation, anti-mutation, anti-tumor, liver protection, hypoglycemia, and anti-inflammation, which confer them a good application prospect. Nevertheless, there is not a comprehensive review of purple sweet potato anthocyanins so far. The extraction, structural characterization, stability, functional activity, application in the food, cosmetics, medicine, and other industries of anthocyanins from purple sweet potato, together with their biotransformation in vitro or by gut microorganism are reviewed in this paper, which provides a reference for further development and utilization of anthocyanins.

Purple potatoes, also known as purple sweet potatoes, are a one-year or perennial herb in the family Convolvulaceae. Their flesh is purple to dark purple. Besides the nutrients of ordinary sweet potatoes, they are also rich in anthocyanins [1]. Purple sweet potato is an important source of dietary fiber, minerals, vitamins, anthocyanins, and so on. It can be used by humans and animals [2]. The recommended intake of cereals and potatoes in *Dietary Guidelines for Chinese Residents* is about 250–400 g/day, of which 50–100 g is potatoes. The daily dietary structure of American residents showed that the average daily anthocyanin intake per capita was about 12.5 mg/day [3]. Therefore, purple sweet potatoes can not only be used as a green food to meet people’s daily intake of cereals and potatoes, but also increase the daily intake of anthocyanins to achieve health effects.

Purple sweet potatoes possess attractive purple-red color, high anthocyanin content, high total phenol content, and high antioxidant activity [4,5,6]. It is reported that the content of anthocyanin in purple sweet potatoes is significantly higher than that in ordinary orange-fleshed sweet potatoes [7], similar to those of anthocyanin crops with the highest yield, such as blueberries, blackberries, cranberries, and grapes [8]. Moreover, purple sweet potato is an important source of natural anthocyanin pigments because of its low cost [9]. Purple sweet potato anthocyanins (PSPAs) is a compound of many varieties of anthocyanins, whose chemical structure is mainly composed of cyanidins and peonidins in the form of monoacylation and diacetylation. Because of its acylation form, PSPAs have high heat resistance and ultraviolet stability, which is their advantage as natural pigments in food additives [10]. At present, PSPAs have been widely used as a food additive in China [11]. PSPAs have a variety of biological activities, including antioxidation [2], anti-inflammation [12], anti-mutation, and anti-tumor [13,14]. In addition, PSPAs can attenuate the damage of dimethylnitrosamineto rat liver [15]. In recent years, a wide variety of researches are focused on PSPAs extraction, purification, structure characterization, stability, biological activity, and biotransformation, but there is not a comprehensive review of PSPAs as far as we know.

## 1. The Extraction Methods of PSPAs

Extraction is a separation technology, which has a certain impact on the extraction rate, quality, and composition of target compounds [16,17]. An effective extraction method can maximize the use of target compounds on the basis of utilizing environmental protection technology [18].

At present, the main extraction method of PSPAs is solvent extraction, it usually uses acid solution to extract step by step. In recent years, with the development of science and technology, some effective processing technologies have been widely used in the extraction of plant active components, such as enzymatic method, ultrasonic method, and pulsed electric field method. These technologies can efficiently destroy cell walls and cell membranes, increase the permeability of tissue cells, shorten the extraction time, enhance the yield of pigments, and improve the quality of products [19]. Different extraction methods can affect the yield, purity, and composition of PSPAs, and at the same time, they may make slight differences in the antioxidant activity of the extracted PSPAs [18]. Therefore, the selection of extraction methods of PSPAs is of great significance for its functional activity and production application.

### 1.1. Solvent Extraction

The solvent extraction methods of PSPAs include water extraction, acidified water extraction, and acidified ethanol extraction [20,21]. The extraction method usually dissolves water-soluble PSPAs in acidic solution by stirring, then extracts the target substance [22]. PSPAs are not easy to be extracted and also not really stable in neutral or weak alkaline solution, so acid solvents are frequently used in the extraction process. Acidic solvents can dissolve water-soluble PSPAs while destroying the cell membrane of purple sweet potato. Liu et al. [23] studied the extraction process of PSPAs by response surface methodology, and the results showed that: when the concentration of ethanol was 25%, the concentration of ammonium sulfate was 22%, with a ratio of solid–liquid as 1:45 and pH 3.3, the optimum extraction rate and partition coefficient of anthocyanins were 90.02% and 19.62, respectively. It is very close to the predicted extraction rate (90.12%) and the predicted partition coefficient (19.73). Fan et al. [10] using acid-ethanol extraction method studied the comprehensive effects of different extraction conditions on the yield and color characteristics of PSPAs (expressed in L*, C*, and H). The results revealed that the yield of PSPAs was the highest (158 mg/100 g dw) when the temperature was 80 °C, the extraction time was 60 min, and the solid–liquid ratio was 1:32.

### 1.2. Enzymatic Extraction

Enzymatic extraction of PSPAs usually uses single enzymes such as cellulase [24] or pectinase [19]. Bao et al. [25] selected α-amylase and pectinase for double enzymatic extraction, which could destroy cell wall and cell membrane of purple sweet potatoes cells more effectively and improve the yield of PSPAs [19]. The results of single factor test and orthogonal test indicated that the optimum extraction conditions of PSPAs were 50 °C in temperature, 1:15 in material–liquid ratio, pH 5.5; 0.25% in α-amylase dosage and 0.10% in pectinase dosage. Under this condition, the yield of PSPAs was 2.71 mg/g FW and the color value was 43.2.

### 1.3. High Voltage Pulse Electric Field Method

The extraction rate of target components largely depends on the cell wall-breaking status of biomaterials. Traditional wall-breaking methods include physical, chemical, and biological methods [26], while high-voltage pulsed electric field is a new technology. In recent years, more and more progress has been made in ameliorating mass transfer operation in food industry [27,28,29]. This process is based on the application of external electric field, which can induce the electroporation of eukaryotic cell membranes and promote the diffusion of solutes. Under moderate electric field (<10 KV/cm) and low energy (<10 KJ/kg), irreversible damage of tissue cells occurs, which accelerates the flow of target components in cells. Puértolas et al. [30] studied the effect of pulsed electric field treatment on the extraction rate of PSPAs, and found that the pulsed electric field treatment of 3.4 kV/cm and 105 us (35 pulses of 3 us) resulted in the highest cell disintegration index at the lowest specific energy requirements (8.92 kJ/kg) and a PSPAs extraction yield of 65.8 mg/100 g fw with water extraction. 

### 1.4. Microwave-Assisted Extraction

Microwave is a kind of instantaneous penetrating heating method, which has several characteristics of strong penetration, good selectivity, and high heating efficiency [31]. Under the action of microwave radiation, plant cell tissues cause molecular motion through ion migration and dipole rotation. This rapid movement can generate friction and produce thermal energy inside the substance, resulting in the rupture of cell walls and tissues, and then the flow out of solutes. Thus, the extraction rate was accelerated and the yield of the product was effectively increased [32,33]. Xu et al. [34] first used response surface methodology to optimize the extraction process of PSPAs under microwave conditions, and achieved good results. Under microwave conditions, the optimum technological conditions for PSPAs are as follows: microwave radiation power 700 W (temperature 70 °C), radiation time 6.37 min, liquid–solid ratio 15:1, hydrochloric acid concentration 0.3%, at which time the total anthocyanin content is 74.66 mg/100 g.

### 1.5. Ultrasound-Assisted Extraction

Zhu et al. [35] applied ultrasound pre-treatment (UAE) technology to extract valuable compounds from purple sweet potatoes, including polyphenols (especially anthocyanins) and proteins. Under the optimum conditions (ultrasonic time 40 min, assisted thermal extraction (80 °C), pH 2.5, ethanol concentration: 58%), the concentrations of polyphenols (3.877 mg/g), anthocyanins (0.293 mg/g), and proteins (0.753 mg/g) were the highest and the specific energy consumption was the lowest (8406 J/mg). The contents of both main anthocyanins and non-anthocyanins in purple sweet potatoes extract extracted by UAE were higher than those extracted by conventional methods. These results showed that UAE had a good impact on the extraction of anthocyanins from purple sweet potatoes.

### 1.6. Accelerated Solvent Extraction

Truong et al. [36] optimized the extraction conditions of different genotypes of purple sweet potatoes by accelerated solvent extraction. Cai et al. [18] studied the extraction efficiency of PSPAs by conventional extraction, ultrasound-assisted extraction, and accelerated solvent extraction, and found that the extraction efficiency of these three methods for PSPAs was as follows: accelerated solvent extraction > ultrasound-assisted extraction > conventional extraction (218–244 mg/100 g FW). Accelerated solvent extraction is a method of extracting anthocyanins from purple sweet potato under high temperature and high pressure in a short time, which is very beneficial to anthocyanins with thermal instability from purple sweet potato. On the contrary, conventional extraction method and ultrasonic-assisted extraction method are long-term, low-temperature extraction methods, whose extracted yield of PSPAs are very low. Regarding the anthocyanins, compared with conventional extraction or ultrasound-assisted extraction, accelerated solvent extraction can extract more diacyl anthocyanins, but less non-acyl and monoacyl anthocyanins. Moreover, anthocyanins extracted by accelerated solvent extraction method are more stable than those of conventional extraction and ultrasound-assisted extraction methods.

## 2. The Structural Characterization of Purple Sweet Potato

Anthocyanins are a class of pigments in plants, which give brilliant colors (like red, blue, or purple) to fruits, vegetables, and flowers. The anthocyanin molecule consists of an aglycone anthocyanin and several sugar moieties [37]. The common anthocyanins in plants are cyanidin, peonidin, gelargonidin, malvidin, delphinidin, petunidin, etc., [38]. Actually, anthocyanins are a class of mixture which has various types and complex structures. It is reported that there are more than 600 kinds of anthocyanins in nature. The structure of these anthocyanins varies according to the types of anthocyanins, the number of sugar molecules and the type of acylation groups. PSPAs are safe, non-toxic, odorless, and rich in resources, and their physicochemical properties are much more stable than those of strawberry anthocyanins, cherry anthocyanins, and grape anthocyanins.

Anthocyanins have many functions including anti-oxidant, anti-tumor, antimutagenic, anti-inflammatory, cardiovascular disease prevention, diabetes prevention, weight loss, and other effects [39,40,41]. Anthocyanins are also considered as a kind of safe and effective food colorant in the food industry [42]. Separation and identification of PSPAs is of great importance for better understanding of their structure and physicochemical properties, and also for enriching their use and developing products with higher added value [43]. At present, the commonly used methods for structure identification of anthocyanins mainly involve ultraviolet-visible spectroscopy, high-performance liquid chromatography-mass spectrometry, and nuclear magnetic resonance.

### 2.1. Ultraviolet-Visible Spectroscopy

Ultraviolet-visible (UV) spectroscopy is a method that can give a preliminary determination of PSPAs structures based on the spectral characteristics. In acidic solution (pH = 3), PSPAs have two characteristic absorption peaks near 275 nm and 530 nm. If the PSPA has a strong absorption peak at about 330 nm, it is acylated anthocyanin, and otherwise it is non-acylated anthocyanin. Among them, if there is an absorption peak at 308–313 nm, it is p-coumaroyl anthocyanin, while it is caffeoyl anthocyanin if the absorption peak is at 326–329 nm. Furthermore, the position and number of acyl groups can be inferred from the ratio of A_440nm_/A_max_ [44]. Chen [45] previously pointed out that PSPAs are mainly acylated anthocyanins according to the strong absorption peaks at 330 nm and 525 nm in the ultraviolet-visible spectra.

### 2.2. High-Performance Liquid Chromatography-Mass Spectrometry

High-performance liquid chromatography-photo-diode array-electrospray ionization-mass spectrometry (HPLC-PDA-ESI-MS^n^) utilizes photo-diode array (PDA) to carry out UV-Vis scanning of each chromatographic peak from high performance liquid chromatography one-by-one first, and at the same time, applying electrospray ionization (ESI) to ionize the chromatographic peaks respectively. Then mass spectrometry (MS) or tandem mass spectrometry (MS^n^) can be used to obtain the molecular weight and fragment ion information of each chromatographic peak, so as to infer the molecular structure of PSPAs [46,47,48,49,50]. Table 1 summarizes the mass spectrum information of glycosides and acylated organic acids commonly found in purple sweet potatoes, thus the structure of PSPAs can be quickly inferred.

Li et al. [47] identified 13 anthocyanins from purple-fleshed sweet potato cultivar Jihei No. 1 by HPLC-PDA-ESI-MS^n^. The main anthocyanins were peonidin and cyanidin with 3-sophoroside-5-glucoside acylated with *p*-hydroxybenzoic acid, ferulic acid, or caffeic acid, among which the content of peonidin3-caffeoylsophoroside-5-glucoside (**1**) was the highest. Besides, the rare delphinidin-3, 5-diglucoside (**2**) was identified in the purple sweet potatoes extract. Zhu et al. [35] identified anthocyanins polyphenols and non-anthocyanin polyphenols in purple sweet potatoes extract by HPLC-DAD-ESI-MS^2^. The results showed that anthocyanins were mainly peonidin3-caffeoyl-*p*-hydroxybenzoylsophoroside-5-glucoside (**3**), peonidin3-(6”-caffeoyl-6”’feruloylsophoroside)-5-glucoside (**4**), cyanidin3-caffeoyl-*p*-hydroxybenzoylsophoroside-5-glucoside (**5**). However, the non-anthocyanin molecules are mainly quinic acid, chlorogenic acid, caffeic acid, and chlorogenic acid-3-glucose. Truong et al. [48] identified 17 anthocyanins from stokes purple sweet potato and purple sweet potato NC 415 by HPLC-DAD/ESI-MS/MS, including **1**, **3**, **5**, cyanidin3-caffeoylsophoroside-5-glucoside (**6**), peonidin-caffeoyl-feruloylsophoroside-5-glucoside (**7**). There are 12 kinds of pigments in Okinawa cultivars, of which three main anthocyanins are (**6)**, cyanidin3-(6”,6”’-dicaffeoylsophoroside)-5-glucoside (**8**), and cyanidin3-(6”-caffeoyl-6”’-feruloylsophoroside)-5-glucoside (**9**) (Figure 1).

### 2.3. Nuclear Magnetic Resonance

Nuclear magnetic resonance (NMR) is an accurate, fast, and high-resolution method for identification, especially for identifying unknown compounds. Nowadays, it has been widely used in chemical analysis from drugs, plants, and microorganisms. NMR is commonly used to identify the structure of substances by multiple one-dimensional and two-dimensional NMR techniques, such as hydrogen spectrum (^1^H NMR), carbon spectrum (^13^C NMR), HMQC, DEPT, HMBC, ^1^H-^1^H COSY, etc. It is more suitable for the structural identification of highly complex acylated PSPAs [51].

At present, many complex chemical structures of PSPAs have been identified by NMR. Zhao et al. [50] used NMR and HPLC-MS^2^ techniques to isolate and identify five PSPAs: 6-*O*-caffeoyl-β-d-fructofuranosyl-(2–1)-α-d-glucopyranoside (**10**), 5-caffeoylquinic acid (**11**), trans-4,5-dicaffeoylquinic acid (**12**), 3,5-dicaffeoylquinic acid (**13**), 4,5-dicaffeoylquinic acid (**14**) (Figure 2), and among them, **10** and **14** was found in purple sweet potatoes for the first time.

Terahara et al. [52] isolated several new acylated anthocyanins from red vinegar fermented from purple sweet potato. The results of MS and NMR emerged that these acylated anthocyanins including cyanidin3-*O*-(6-*O*-(E)-caffeoyl-(2-*O-*(6-*O*-(*E*)-feruloyl)-β-d-glucopyranosyl)-β-d-glucopyranoside) (**15**), peonidin3-*O*-(6-*O*-(*E*)-caffeoyl-(2-*O*-(6-*O*-acyl)-d-glucopyranosyl)-β-d-glucopyranosides) (**16**), peonidin3-*O*-(6-*O*-p-hydroxybenzoyl-(2-*O*-(6-*O*-acyl)-d-glucopyranosyl-β-d-glucopyranosides) (**17**), and peonidin3-*O*-(6-*O*-(*E*)-feruloyl-(2-*O*-(6-*O*-acyl)-d-glucopyranosyl-β-d-glucopyranosides) (**18**) (Figure 3). Six anthocyanin components were identified from Japanese purple sweet potato (*Ipomoea batatas* cv. Yamagawamurasaki) by nuclear magnetic resonance, mainly composed of the derivatives of cyanidin and peonidin [53,54].

## 3. The Stability of PSPAs

As a special type of sweet potato, purple sweet potato is rich in natural anthocyanins with various physiological activities, good quality, and brilliant appearance. In recent years, it has attracted many scholars’ attention [55,56,57], and has become a natural red pigment resource with healthcare functions. Much progress has been made in the researches of molecular structure and components of PSPAs, and revealed that the main components of PSPAs are acyl-containing anthocyanins and methyl anthocyanins. PSPAs possess stronger dyeing power to beverages, pastries, cold drinks, and other foods, and better stability than other anthocyanin pigments from blackberry, grape peel, etc., [58,59]. Many researches have studied the stability and physicochemical properties of PSPAs, which provided a theoretical basis for their processing and utilization.

### 3.1. Effect of Chemical Structure on Stability

Numerous studies have shown that the chemical structure of PSPAs has a great influence on their overall stability. Because PSPAs contain a large number of acylated organic acid groups, the formation of steric hindrance prevents the C_2_ position of phenyl benzopyran cation O^+^ in parent nucleus from being attacked by water molecules, which is conducive to maintain the original structure and is difficult to form colorless pseudobase and chalcone structure [44]. Therefore, PSPAs are more stable than other anthocyanins.

Different from anthocyanins in other berries, PSPAs mainly exist in the acylation form [60,61]. Acylation with various phenolic acids gives PSPAs some unique properties and has certain advantages in pH resistance, heat resistance, photosensitivity, and overall stability [7].

### 3.2. Effect of Processing Method on Stability

Xu et al. [7] authenticated 12 acylated anthocyanins from purple sweet potatoes P40. After four cooking conditions like steaming, high pressure cooking, microwaving and frying, the total amount of PSPAs decreased by 8–16%, while baking had no significant effect on the total amount of PSPAs. In addition, they also found mono-acylated anthocyanins exhibited better thermal stability than di-acylated and non-acylated anthocyanins, of which cyanidin3-*p*-hydroxybenzoylsophoroside-5-glucoside (**19**) had the highest thermal stability. After microwave treatment, the content of **19** in purple sweet potatoes P40 increased significantly from 121 mg/100g to 462 mg/100g, and the content of peonidin3-p-hydroxybenzoylsophoroside-5-glucoside (**20**) (Figure 4) increased from 19 mg/100g to 87 mg/100g, suggesting that some processing methods may release more anthocyanin monomers from purple sweet potatoes.

Qi et al. [62] identified 9 PSPAs from American black potatoes and 17 PSPAs from Yunnan purple sweet potatoes. After five common Chinese cooking processes including steaming, boiling, roasting, microwaving and low-temperature frying, PSPAs were lost in different degrees. The loss rate was microwave < steaming < boiling < low-temperature frying < roasting. More specifically, the loss rate of PSPAs after microwaving was 13–14%, but the loss rate after roasting was 62–6%.

The studies from Kim et al. [63] showed that the total content of PSPAs in Korean purple sweet potatoes decreased by nearly half after steaming, and the content of all monomer anthocyanins were declined in various degrees. However, the loss rate of PSPAs after baking was very low, because baking just reduced the content of acylated anthocyanins but increased non-acylated anthocyanins content.

### 3.3. Effect of Other Factors on Stability

In addition to the above factors, pH, light, metal ions and other factors may also have a certain impact on the stability of PSPAs. The results of Chen [45] showed that the retention rates of PSPAs in dark, indoor light, outdoor light, and ultraviolet light conditions were 89.90%, 82.22%, 71.98%, and 50.64%, respectively, so they indicated that PSPAs had poor photostability, especially sensitive to ultraviolet light. With the range from pH 2 to pH 11, the color of PSPAs solution gradually changed from dark red to blue, and the maximum absorption wavelength showed a trend shift toward the color blue. Besides Al^3+^ has a slight color protection effect on PSPAs, Fe^3+^, Cu^2+^, Zn^2+^, Pb^2+^ and other metal ions will reduce the stability of PSPAs.

## 4. Functional Activity of PSPAs

### 4.1. Antioxidant Activity

Humans produce various free radicals in the metabolic process, and excessive free radicals can lead to oxidation of lipids, proteins, DNA, RNA, and sugar, which are associated with cancer, Alzheimer’s disease, Parkinson’s disease, autoimmune deficiency, diabetes, obesity, and other diseases [64,65]. Teow et al. [2] used ORAC, DPPH, and ABTS to study the free radical scavenging ability of 19 kinds of sweet potatoes with white, light yellow, yellow, orange, and purple sarcocarp, the results showed that purple sweet potato had the strongest antioxidant ability. This is mainly because PSPAs can offer protection against the inflammatory progression from oxidative stress and induce a decline in the various oxidative stress markers. The mechanism is meant to reduce the enzymes which encourage the proliferative process, as well as to protect against the decline in the nitric oxide level [66]. Kano et al. [67] studied the extraction of Ayamurasaki cultivar and found that its scavenging ability of DPPH free radical was higher than that of purple cabbage, grape peel, elderberry, and purple corn extract, and the scavenging ability of eight PSPAs on DPPH free radical was better than that of ascorbic acid.

### 4.2. Antimutagenic and Anti-Tumor Activities

Yoshimoto et al. [13] tested the anti-mutation ability of four different PSPAs including cyanidin3-sophoroside-5-glucoside (**21**), cyanidin3-(6, 6′–caffeylferulylsophoroside)-5-glucoside (**22**), peonidin3-sophoroside-5-glucoside (**23**), peonidin3-(6, 6′–caffeylferulyl-sophoroside)-5-glucoside (**24**) (Figure 5) with *salmonella typhimurium TA98*. The results showed that the anti-mutation ability of cyanidin type anthocyanins was stronger than that of peondin type anthocyanins, and the anti-mutation ability of peondin type anthocyanins decreased significantly after diacylation, because the cathecol structure plays an important role in the strong antimutagenicity of anthocyanin pigments. Thus, the purple sweet potato varieties with high content of cyanidin-type PSPAs would be superior as physiologically functional foods.

The study of Zhao et al. [14] showed that PSPAs inhibited the growth of implanted S180 tumor cells in mice. In the negative control group, the mortality of tumor cells was both less than 5% after five days and one week of S180 tumor cells implantation. After S180 tumor cells were implanted for one week, the average weight of mice in the negative group was lower than that in the PSPAs group. Low dose, medium dose, and high dose of PSPAs all inhibit tumor growth in mice. When PSPAs concentration reached 500 and 1000 mg/kg BW, the inhibition rate was 43.28% and 68.03% respectively, and the apoptosis rate was proportional to the PSPAs concentration. Because PSPAs consumption could elevate glutathione peroxidase and superoxide dismutase levels and lower malondialdehyde levels; the inhibition of tumor growth may be achieved through enhancement of antioxidant activities. 

### 4.3. Liver Protection

Carbon tetrachloride can damage the liver, and its metabolite trichloromethyl free radical (CCl_3_•) can cause acute hepatitis and induce the liver to release large amounts of glutamine transaminase (GOT) and glutein transaminase (GPT). Suda et al. [68] conducted animal experiments on rats, and found that feeding rats with carbon tetrachloride would cause acute hepatitis together with the increase of GOT and GPT in serum, while the increase of GOT and GPT in serum of mice simultaneously fed with purple sweet potato beverage was significantly inhibited. Then they conducted human clinical experiments on 45 volunteers, and divided them into two groups according to five years of abnormal liver function into those with less than five years and those with more than five years. The experimental group was provided with 120 mL purple sweet potato juice every day for 44 consecutive days, and the results showed that all indexes of liver function decreased by 20% on average in patients with high liver function and disease course within five years [69]. Later, human experiments on another group of volunteers indicated that purple sweet potato beverage could also reduce the level of hepatitis index enzymes in serum such as aspartate aminotransferase (AST), alanine aminotransferase (ALT), and especially γ-glutamyl transferase (GGT), because PSPAs can alleviate oxidative stress [70]. In addition, other studies have shown that purple sweet potato also has a significant inhibitory effect on liver injury caused by cholesterol and d-galactose [12,71]. 

Hwang et al. [72] found that in HepG2 cell model and rat experiments, PSPAs can effectively reduce the oxidative damage induced by *tert*-butyl hydroperoxide, and significantly reduce the incidence of liver lesions. Zhang et al. [73] showed that PSPAs could effectively inhibit the production of reactive oxygen species in mice, reduce the level of reactive oxygen species, and effectively inhibit high-fat diet (HFD)-induced obesity and related liver fat accumulation by activating adenosine-monophosphate activated protein kinase (AMPK) signaling pathway and adipogenesis, thereby reducing the sensitivity of oxidative stress and liver insulin resistance.

Cai et al. [74] isolated 12 kinds of PSPAs from purple sweet potato and studied their effects on liver injury caused by alcohol. Researches have shown that alcoholic beverages with low, medium, and high doses of PSPAs have mutually supporting effects on major liver function indicators, liver histological changes and oxidation status of mice, and medium doses of PSPAs have obvious protective effects on alcoholic liver injury mice by reducing the release of ALT in liver cells. However, large doses of PSPAs (300 mg/kg·BW·d) may promote alcohol-induced liver injury by promoting oxidation, which depends on MDA and GSH levels. Therefore, PSPAs at appropriate dose can be used as another way to reduce alcoholic liver injury.

### 4.4. Other Function

In addition to the above functional activities, PSPAs also has the functions of anti-inflammation, anti-cancer, hypoglycemic, and improving intestinal microecology.

Wang et al. [75] found that PSPAs could significantly reduce the expression of tumor necrosis factor-alpha (TNF-α), interleukin-1beta (IL-1β) and interleukin-6 (IL-6) in brain cells of rats with accumulative lipopolysaccharide, thus to alleviate acute inflammation of brain.

Li et al. [76] extracted concentrated anthocyanin from purple sweet potato and intervened BIU87 cells with anthocyanin of different concentrations (100, 200, 400, 800 μg/mL) for 48 h. The inhibitory effect of anthocyanin on the proliferation of BIU87 cells was detected by cell morphology observation and cell counting kit (CCK-8 method), and the apoptosis rate was analyzed by flow cytometry (FCM). The results showed that with the increase of anthocyanin concentration, the number of BIU87 cells decreased, the cell volume lessened, the cell space increased, the cell adhesion became worse, and the cell morphology was deformed. CCK-8 method demonstrated that anthocyanins of different concentrations had a significant inhibitory effect on the proliferation of BIU87 cells. The absorbance values of BIU87 cells treated with 100, 200, 400, and 800 μg/mL anthocyanins for 48 h were 1.24 ± 0.07, 1.15 ± 0.11, 0.90 ± 0.08, and 0.56 ± 0.09, respectively, which had statistical significance compared with the control group (all *P* < 0.05). The results of FCM exhibited that the apoptosis rates were 7.31%, 11.11%, 25.96%, 36.28% in the 100, 200, 400, and 800 μg/mL anthocyanin groups, and the apoptosis rates gradually enhanced with the increase of anthocyanin concentration. Therefore, PSPAs inhibit the proliferation of bladder cancer BIU87 cells by inducing apoptosis, and it is dose-dependent.

Matsui et al. [77] gave maltose of 2 g/kg BW first and then diacylated PSPAs of 100 mg/kg BW orally to male mice at eight weeks old, and the results showed that the blood glucose concentration in mice obviously decreased by 16.5% after 30 min, and the diacylated anthocyanin reduced the blood glucose level by inhibiting the activity of α-glucosidase.

Zhang et al. [78] prepared PSPAs by column chromatography, and analyzed the effect of PSPAs on intestinal flora by observing the changes of bacterial population and short-chain fatty acid (SCFA) concentration at different time points. It was found that PSPAs, including corresponding monomers, could evidently increase the population number of *Bifidobacterium* and *Lactobacillus/Enterococcus spp* and the concentration of SCFA. Therefore, the intake of cyanidin-rich purple sweet potatoes may benefit to intestinal microecology and host health. 

## 5. Application of PSPAs

In recent years, with the questioning for the safety of synthetic pigments, natural pigments are widely favored by consumers. Because of high safety, bright color, and physiological functions, PSPAs are widely used in ice cream, plant drinks, candy, dairy drinks, aquatic products, cereal products, and other food industries, as well as cosmetics industries such as rouge and lipstick, and can also be used for coloring medical tablets [79].

### 5.1. Application of PSPAs in Food Industry 

PSPAs have high heat resistance and stability because of its acylation form. In addition to being used as a natural pigment, it can also be processed into concentrate, paste, and flour. In Japan, many foods containing processed purple sweet potatoes, such as noodles, breads, jams, crisps, candies, beverages, and alcoholic beverages, are widely manufactured and sold [80]. Furthermore, the study of Bassa et al. [81] showed that PSPAs were more stable in beverages than other commercial pigments and blackberry pigments.

Tensiska et al. [82] added the embedded PSPAs to jelly beverage and stored them in a refrigerator without light, which can not only maintain the stability of anthocyanin quantity, color intensity, and acidity (pH), but also extend the shelf-life of jelly drinks. After 30 days of storage, the total anthocyanin content of the jelly beverages decreased the smallest (46.03%) and the red intensity declined by 3.19%, while the shelf-life of jelly beverages reached to about 200 days.

### 5.2. Application of PSPAs in Cosmetics Industry

PSPAs have obvious activities such as scavenging free radicals and anti-oxidation, and can also inhibit the generation of oxygen free radicals in ultraviolet radiation. Therefore, skin care products with PSPAs added have certain effects on improving skin inflammation, anti-oxidation, etc. Moreover, PSPAs can replace synthetic pigments which are widely utilized in cosmetics such as lipstick, rouge, and shampoo [83], which can effectively improve the safety of cosmetics industry.

### 5.3. Application of PSPAs in Pharmaceutical Industry

In order to make drugs easier to identify and distinguish, synthetic pigments such as mustard red, carmine, and indigo need to be added in the process of drug production. Long-term use of drugs with synthetic pigments can do harm to human body. Therefore, safe and non-toxic PSPAs can be used to produce colored tablets instead of synthetic pigments [84].

## 6. Biotransformation of PSPAs

More and more evidences have proved that anthocyanins are beneficial to human health. Different anthocyanins have great differences in stability, absorption, metabolism, and elimination. These differences are important in determining the sites of their impact on health, microbiome interactions, and chemical degradation of their transformation. A research from Kano and co-workers found that PSPAs showed stronger DPPH radical-scavenging activity than anthocyanins from red cabbage, grape skin, elderberry, or purple corn. Moreover, DPPH radical-scavenging activity in urine was increased in PSPAs-injected rats, because the elevation of plasma transaminase activities was depressed. Cyanidin 3-*O*-(2-*O*-(6-*O*-(*E*)-caffeoyl-β-d-glucopyranocyl)-β-d-glucopyranoide)-5-*O*-β-d-glucopyranoside (**25**) and peonidin 3-*O*-(2-*O*-(6-*O*-(*E*)-caffeoyl-β-d-glucopyranocyl)-β-d-glucopyranoide)-5-*O*-β-d-glucopyranoside (**26**) (Figure 6) which were detected in the plasma can protect low density lipoprotein from oxidation at a physiological concentration [67]. Actually, the orally administered PSPAs are directly absorbed and excreted, but the metabolic products of the PSPAs absorbed and the mechanism of action of PSPAs remain to be investigated.

Cyanidin-3-glucoside (**27**) (Figure 7) and peonidin-3-glucoside (**28**) (Figure 8) are two anthocyanins widely studied at present, and they are the main components in fruits such as blackberry and strawberry [85]. **27** was also widely found in purple sweet potato. Chen et al. [86] studied bacterial-dependent metabolism of **27** under simulated large intestine. It was found that **27** derives chalcone pseudobases and multiform anthocyanins as well as their methylated metabolites, but these substances cannot be absorbed directly by gut microbiota, so they need to be further metabolized into absorbable compounds under the action of gut microbiota. In the process of **27** transformation, protocatechuic acid (**29**) was the first substance detected. In the detection of **29**, researchers also found the existence of 2,4,6-trihydroxybenzaldehyde (**30**). Surprisingly, after a period of time, two substances, *p*-coumaric acid (**31**) and vanillin acid (**32**) appeared. With the progress of transformation, the presence of *p*-hydroxybenzoic acid (**33**) was found after **27** disappeared. The specific transformation steps are in Figure 7. Therefore, the PSPAs can be transformed into specific metabolites that exert a protective effect in the host physiology after intake. Generally, glycosylated PSPAs can be rapidly metabolized by the gut microflora, but some other metabolic processes can also result in their disappearance, for example, some PSPAs metabolites can react with different groups in some macromolecules and then be degraded into some unexpected metabolites.

Aura et al. [87] identified **27** and cyanidin aglycone as an intermediate metabolite of cyanidin-3-rutinoside (**34**) (Figure 9) after fermentation with a 1% faecal flora, which confirmed the deglycosylation of anthocyanins by the gut flora and the presence of deconjugating enzymes. In addition, **27** is deglycosylated. **29** was the main metabolite, and two unrecognized metabolites were formed. These metabolites did not exist in the fecal background or in the samples incubated with inactive bacteria, indicating that anthocyanins were transformed by intestinal microflora, and the metabolism of anthocyanins by bacteria included the cleavage of glycosidic linkages and breakdown of the anthocyanidin heterocycle. Therefore, it is not adequate to predict the health effects of PSPAs by their intact structures, because they can be transformed by bacteria in the colon to smaller phenolic compounds or conjugates of the aglycone, of which health implications are poorly characterized.

A recent study by Czank et al. [88] using ^13^C-labelled anthocyanins in humans found that the relative bioavailability of **27** was 12.38 ± 1.38% on average based on the recovered ^13^C dose, from which 0.18 ± 0.11% of the ^13^C dose was recovered from blood, 5.37 ± 0.67% from urine, and 6.91 ± 1.59% from breath, whereas 32.13 ± 6.13% of the^13^C was found in feces. These data indicate that the relative bioavailability of anthocyanins is 2.5% to 18.5% as that of other flavonoid subclasses, such as flavan-3-ols and flavones. The rapid appearance of **27** degradation products and their phase II conjugates within the serum suggests that some degradation may occur in the small intestine (before or after absorption), so the cleavage of the anthocyanin C-ring may not require the action of colonic microflora. Subsequently, Flores et al. [89] investigated the release of anthocyanins and the formation of phenolic microbial metabolites after anaerobic batch-culture fermentation of cyclodextrin (CD) embedded anthocyanins with gut bacteria, and found that the maximum released concentration was 54%, 61%, and 85% for delphinidin-3-glucoside (**35**), **27** and malvidin-3-glucoside (**36**) (Figure 10), respectively. The difference of anthocyanin release rate may be related to host–guest interaction which is a function of two key factors. One involves space requirements and depends on the relative size of CD and the size and physicochemical properties of the molecules involved. The second is the thermodynamic interaction between different components of the system. The inclusion complexes of these host-guest systems occur through various interactions, such as hydrogen bonding, van der Waals interaction, hydrophobic interaction, and electrostatic attraction. Different structures of anthocyanins enable them to interact differently with β-CD. The effect of bacterial metabolism resulted in the rupture of glycosidic linkages and cleavage of the anthocyanin heterocycle ring. LC-MS results revealed that gallic acid, ferulic acid, and syringic acid were the main phenolic acids detected when **35**, **27,** and **36** were respectively fermented.

Kubow et al. [90] first digested purple sweet potato (PFSP) PM09.812 and PM09.960 through the dynamic human gastrointestinal model (GI) simulating intestinal digestion conditions, and studied the biotransformation of anthocyanins. The results showed that the release and degradation of anthocyanins varied with time. After 24 h, more anthocyanins were detected in small intestinal vessels compared with other vessels added PM09.960, while more anthocyanins were detected in ascending colon vessels entering PM09.812. In the small intestinal vessels added with PM09.960 and the ascending colon vessels added with PM09.812, the antioxidant capacity of ferric was increased, which corresponds to the majority of anthocyanins presented at each addition. These results suggest that the release of anthocyanins in sweet potato food matrix is an additive-dependent process under the simulated digestion of stomach, intestine, and colon. There was a significant loss of anthocyanin species detected during the digestion in the intestine and colon, unlike the addition of sweet potatoes. These findings suggest that variations in other food components affect the release of anthocyanins during digestion, which in turn affects their antioxidant capacity in the small intestine.

The cell line of purple sweet potato can be suspension cultured in vitro. When a high-anthocyanin accumulating cell line generated from the storage root of sweet potato (*Ipomoea batatas* L.), cv. Ayamurasakiwas cultured in modified Murashige and Skoog medium, the total amount of phenolic compounds including chlorogenic acid and caffeic acid increased three-fold over four days and remained at the same level over the whole growth period, while relative concentrations of non-acylated anthocyanins were drastically decreased [86]. This is mainly because suspension culture changes the growing environment and metabolic pathways of purple sweet potato cells.

## 7. Application Prospects

With the enhancement of people’s awareness of food safety, the safety of edible pigments has attracted widespread attention. Countries have also introduced restrictions on the use of synthetic pigments in food, such as the United States has decreased the kinds of used synthetic pigments from the original 24 to current 7. China has also decreased them from original 49 to current 17. As a safe food pigment, natural pigment is increasingly favored by consumers, so it is imperative to replace synthetic pigment with natural pigment. According to relevant statistics, the growth rate of natural pigments in the international market has been maintained at more than 10% in recent years. At present, the price of natural pigments on the market is very high, especially in Japan, the United States, Germany, and other developed countries. Therefore, it is urgent to find a natural pigment resource with high quality, high yield, and low price [84].

As a natural edible pigment, PSPAs is safe, non-toxic, odorless, bright in color [14], and they have the advantages of high yield, low cost, rich nutrition, better light and heat stability, and certain pharmacological effects [80]. It is difficult to find plants that can compete with high anthocyanin purple sweet potato in nature. As a new natural pigment resource, it has broad application prospects in food, cosmetics, and medicine industries.

## Figures and Tables

**Figure 1 molecules-24-03816-f001:**
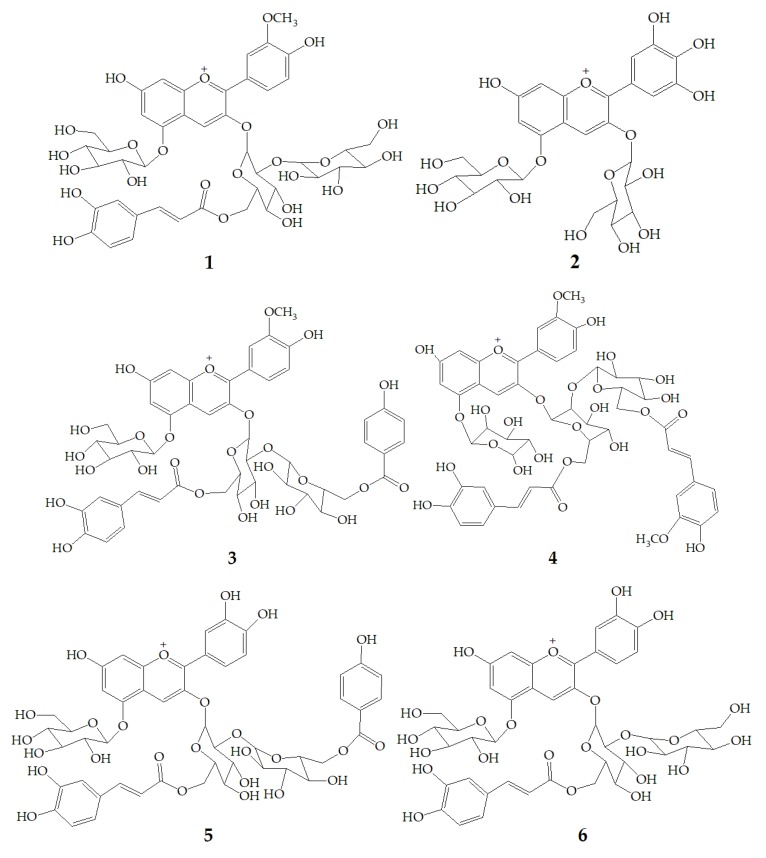
Chemical structures of compound **1**–**9.**

**Figure 2 molecules-24-03816-f002:**
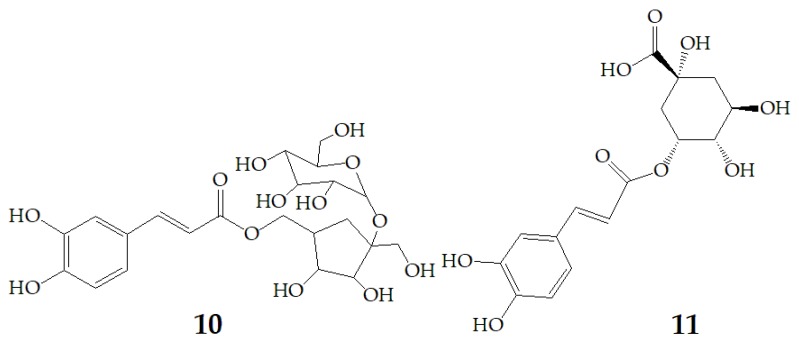
Chemical structures of compound **10**–**14**.

**Figure 3 molecules-24-03816-f003:**
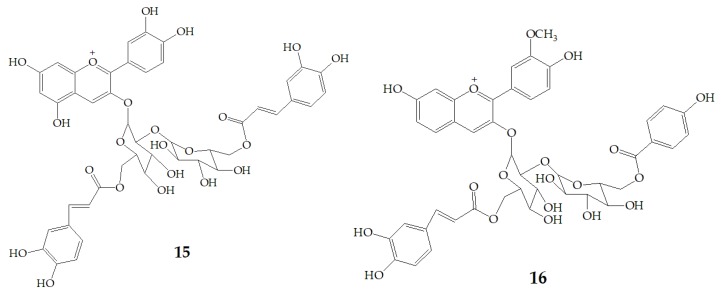
Chemical structures of compound **15**–**18**.

**Figure 4 molecules-24-03816-f004:**
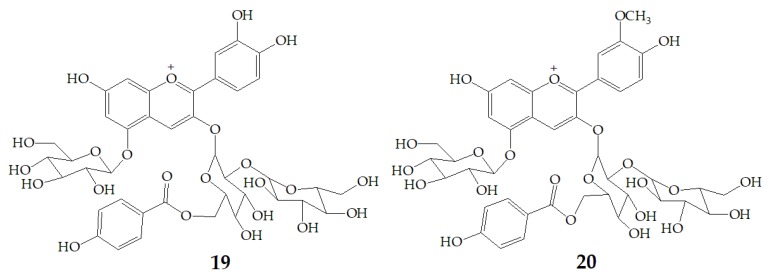
Chemical structures of compound **19**, **20**.

**Figure 5 molecules-24-03816-f005:**
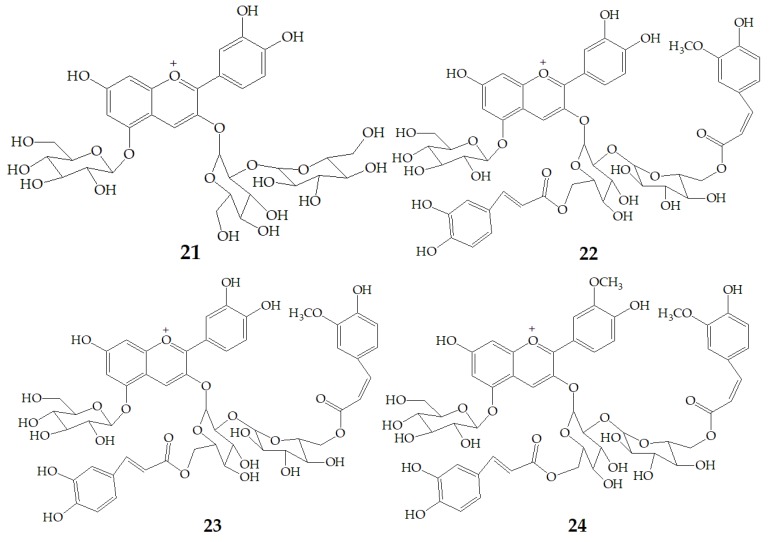
Chemical structures of compound **21**–**24**.

**Figure 6 molecules-24-03816-f006:**
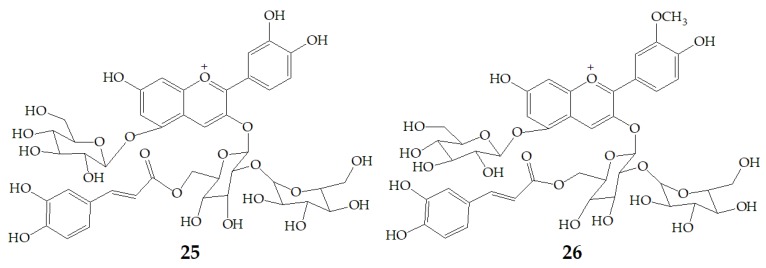
Chemical structures of compound **25**, **26**.

**Figure 7 molecules-24-03816-f007:**
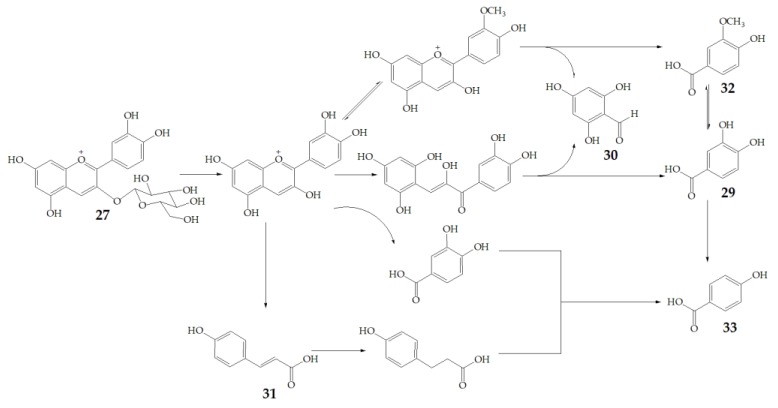
Biotransformation pathway of **27**.

**Figure 8 molecules-24-03816-f008:**
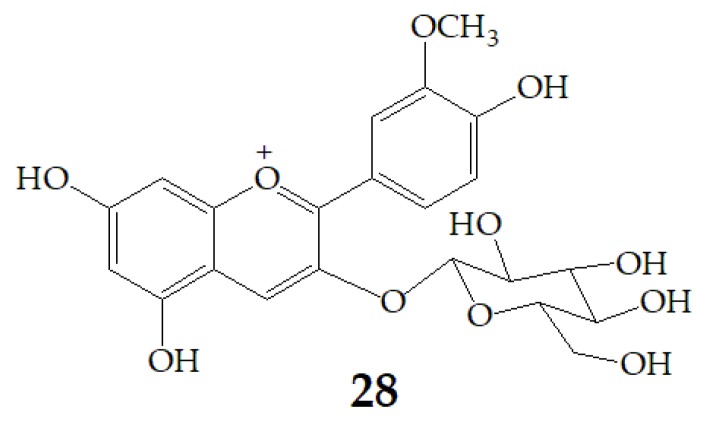
Chemical structures of compound **28**.

**Figure 9 molecules-24-03816-f009:**
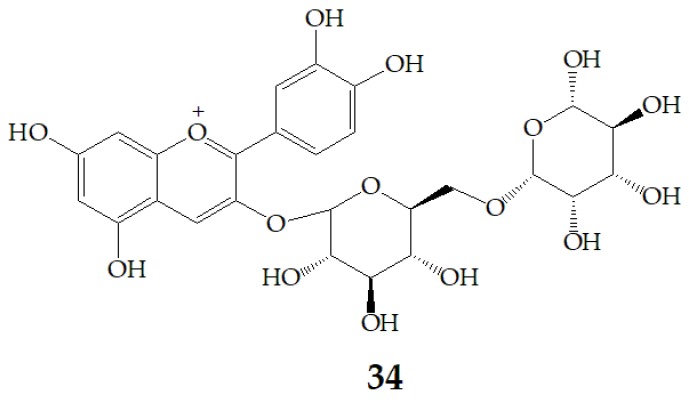
Chemical structures of compound **34**.

**Figure 10 molecules-24-03816-f010:**
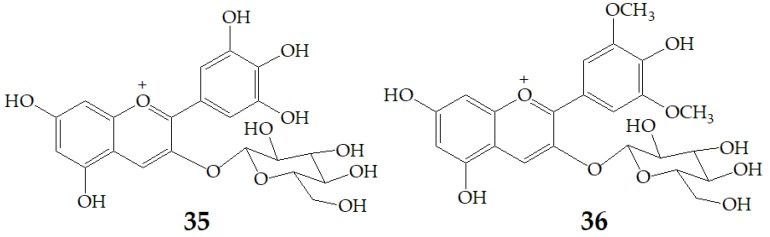
Chemical structures of compound **35**, **36**.

**Table 1 molecules-24-03816-t001:** Structural information of glycosides and acyl substrates in common purple sweet potato anthocyanins (PSPAs).

Name	Structure	Fragment Ion Peak
Glucose	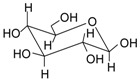	162
Sophorose	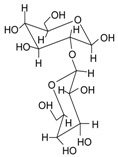	324
Ferulic acid	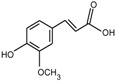	176
Caffeic acid	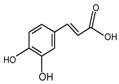	162
P-coumaric acid	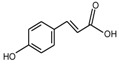	146
Sinapic acid	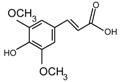	206
Gallic acid	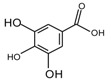	152
Malic acid	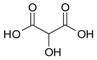	116
Tartaric acid	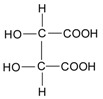	132
Oxalate	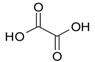	72
Propanedioic acid	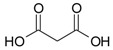	86
Succinic acid	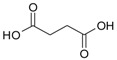	100
P-hydroxybenzoic acid	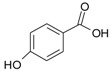	120

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
