# Peer review of "Research Advances of Purple Sweet Potato Anthocyanins: Extraction, Identification, Stability, Bioactivity, Application, and Biotransformation"

_molecules, 2019, doi:10.3390/molecules24213816_

Round 1

Reviewer 1 Report

The manuscript consists of a literature survey on Antocyanins obtained from Purple Sweet Potatoes. It is original and interesting contribution. Majority of the relevant reports from the past two decades are included as reference citations. Overall the manuscript is very well-designed, with precise sections and subsections. The relevant structures are aesthetically presented. Also, the practical applications and the future prospects for the compounds are properly concluded.

Certainly there is a niche for such review article. Therefore I recommend the acceptance of this manuscript for publication in current form. Only a minor spell and grammar check is recommended. 

Author Response

Responses to the first reviewer's questions

Question 1: - a minor spell and grammar should be rechecked

Answer:Thank you for your comments and accepting our manuscript. As for the questions about our spelling and grammatical errors that you mentioned, we have tried our best to revise them and marked in red. Finally, thank you again for your guidance.

Reviewer 2 Report

Dear Authors,

After the review process, I have several comments:

you should clearly define the aim of the paper in abstract and introduction sections; you should expand the discussion about the mechanisms of the interactions of nutraceuticals from sweet potato with microbiota (you could find an example in https://doi.org/10.3389/fphar.2019.00492). I recommend them to insert a figure to present the process; you should keep only the chemical structure of essential compounds for the aim of the paper; yo should comment bioavailability conditions based on the microbiota interactions and the significance of modulation process; you should insert their comments about bioactivity based on the consumption compared to other potatoes species; you should increase the number of references from the last five years, part of them are too old.

Best regards!

Author Response

Responses to the second reviewer's questions

Question 1: - the aim of the paper in abstract and introduction sections should be clearly defined

Answer:Thank you. We have clearly defined the aim of the paper in abstract and introduction sections of the revised manuscript and marked them in yellow background (Line 21-22, Line 25-26, Line 51-53).

Question 2: - expand the discussion about the mechanisms of the interactions of nutraceuticals from sweet potato with microbiota and insert a figure to present the process

Answer:Good suggestion. We have inserted some discussion about the mechanisms of the interactions of nutraceuticals in the revised manuscript and marked them in yellow background (Line 305-308, Line 318-321, Line 424-426, Line 456-458).

Question 3: - keep only the chemical structure of essential compounds for the aim of the paper

Answer:Thank you for your suggestion. The chemical structures in the manuscript are very important purple sweet potato anthocyanins. They will help readers understand the structure of purple sweet potato anthocyanins.

Question 4: - comment bioavailability conditions based on the microbiota interactions and the significance of modulation process

- insert their comments about bioactivity based on the consumption compared to other potatoes species

Answer:Very good suggestions. We have added some comments about PSPAs bioavailability based on the microbiota interactions (Line 440-444, Line 450-451, Line 465-467, Line 474-480) or based on the consumption compared to other potatoes species (Line 305-308) and the significance of modulation process (Line 291-295, Line 318-321, Line 336-337, Line 342-345) in the revised manuscript and marked them in yellow background.

Question 5 - increase the number of references from the last five years

Answer:Thank you. We have added some new references in the revised manuscript.

Round 2

Reviewer 2 Report

Dear Authors,

I do not have any supplementary comments.

Best regards!